# SEG4SEG: Identifying Systematic Failure Modes in Segmentation by Subgroup Discovery Methods

**Nina Weng**[1] (iD)                                                                NINWE@DTU.DK
**Eike Petersen**[2,3]                                        EIKE.PETERSEN@MEVIS.FRAUNHOFER.DE
**Alceu Bissoto**[4,5]                                                   ALCEU.BISSOTO@UNIBE.CH
**Susu Sun**[6]                                                   SUSU.SUN@UNI-TUEBINGEN.DE
**Lisa M. Koch**[4,5]                                                       LISA.KOCH@UNIBE.CH
**Aasa Feragen**[1]                                                              AFHAR@DTU.DK
**Siavash Bigdeli**[1]                                                            SARBI@DTU.DK
**Christian F. Baumgartner**[7]                        CHRISTIAN.BAUMGARTNER@UNILU.CH

[1] *Technical University of Denmark, Kongens Lyngby, Denmark*

[2] *Fraunhofer Institute for Digital Medicine MEVIS, Bremen, Germany*

[3] *Hannover Medical School, Institute for Diagnostic and Interventional Radiology, Hanover, Germany*

[4] *Department of Diabetes, Endocrinology, Nutritional Medicine and Metabolism UDEM, Inselspital, Bern University Hospital, University of Bern, Switzerland*

[5] *Department of Digital Medicine, University of Bern, Switzerland*

[6] *Cluster of Excellence: Machine Learning - New Perspectives for Science, University of Tübingen, Germany*

[7] *Faculty of Health Sciences and Medicine, University of Lucerne, Switzerland*

**Editors:** Accepted for publication at MIDL 2026

## Abstract

Deep learning models for medical image segmentation can achieve high overall performance but fail systematically on critical subgroups. While Slice Discovery Methods (SDM) have shown promise in revealing classification failures, their effectiveness for segmentation remains unexplored. Moreover, although various systematic failures have been reported in segmentation tasks, no prior work has systematically categorized them. In this work, we address both gaps. First, we categorize potential sources of systematic errors in medical image segmentation. Second, we empirically investigate whether SDMs can identify problematic slices in each of those categories without manual annotations. Our evaluation covers four controlled failure types and two real-world failure cases, using medical imaging datasets and explicit success criteria for SDM evaluation. Our experiments show that SDMs adapted for segmentation can identify systematic errors, demonstrating their potential for failure analysis in medical imaging. Our code is publicly available at nina-weng.github.io/seg4seg.github.io.

**Keywords:** Subgroup discovery, segmentation, systematic error, shortcut learning

## 1. Introduction

Understanding why machine learning models fail in high-stakes domains such as medical imaging can reveal important root causes of model failures. As a motivating example, consider the case of pneumothorax classification from chest X-rays, for which Larrazabal et al.

(2020) documented a significant gender performance gap. An explanation – and, thus, a potential solution – of this phenomenon remained elusive, however. Initial hypotheses centered on underrepresentation and biological factors (e.g., breast shadows), yet both were systematically ruled out (Weng et al., 2023). Finally, Olesen et al. (2024) applied Slice Discovery Methods (SDMs) to investigate this problem, revealing that the model had learned a shortcut based on the presence of chest drains in the images. Chest drain prevalence differs systematically between genders, thereby linking the shortcut to the previously unexplained gender performance disparity and, finally, providing a root cause explanation.[1]

Slice discovery methods (SDMs) are unsupervised or semi-supervised clustering methods that aim to identify semantically coherent clusters (slices) of input data that differ in model performance, thereby aiding the discovery of model failure modes (Eyuboglu et al., 2022; Bissoto et al., 2025; Olesen et al., 2024). SDMs have been predominantly applied in the realm of image classification, and a successful application to segmentation tasks has not been demonstrated. Similar to classification models, segmentation models have also been shown to suffer from systematic subgroup performance differences, however. Puyol-Antón et al. (2022) demonstrated racial bias in DL-based cine CMR segmentation, Li et al. (2024a) showed demographic performance gaps in SAM-based abdominal organ CT segmentation, Ioannou et al. (2022) found significant sex and racial performance gaps in brain MR segmentation, and many further studies have presented similar findings (Li et al., 2024b; Čevora et al., 2024; Dou et al., 2024). Yet most such studies do not investigate the root causes of these observed disparities – which, as our motivating example illustrates, is crucial for successfully addressing these failure modes and making models more robust. Motivated by this gap, we here extend the slice discovery paradigm to segmentation tasks.

Significant gaps remain in applying SDMs to segmentation tasks for medical imaging:

**Issue 1: Lack of focus on segmentation tasks.** Most existing SDM frameworks target classification with single-label predictions. Segmentation operates on pixel-level annotations, fundamentally expanding the failure space: models can fail through boundary errors, spatial shortcuts, or texture biases that are absent in classification. While recent work shows shortcut learning occurs at both sample and pixel levels (Lin et al., 2024), a systematic investigation of SDMs for segmentation failures remains absent.

**Issue 2: Absence of success criteria for slice discovery results.** To use slice discovery in practice, a way to evaluate whether the discovered slices genuinely capture systematic errors is needed. However, existing SDMs lack clear evaluation criteria, making it challenging to determine whether an SDM actually works.

**Issue 3: Lack of taxonomy for segmentation failures.** Unlike classification, segmentation lacks a systematic taxonomy of failure types. This gap hinders both the development of targeted SDM approaches and the interpretation of discovered slices.

Accordingly, the **core focus and contributions** of our work are:

1. We adapt SDMs to segmentation for the first time through our proposed SEG4SEG pipeline (Fig. 1) and analyze embedding variants tailored to segmentation tasks.
2. We redefine evaluation metrics and propose principled criteria for assessing slice discovery quality.

---

1. The pneumothorax–chest drain shortcut had been documented earlier, but the connection to gender performance gaps was unknown (Oakden-Rayner et al., 2020; Jiménez-Sánchez et al., 2023).

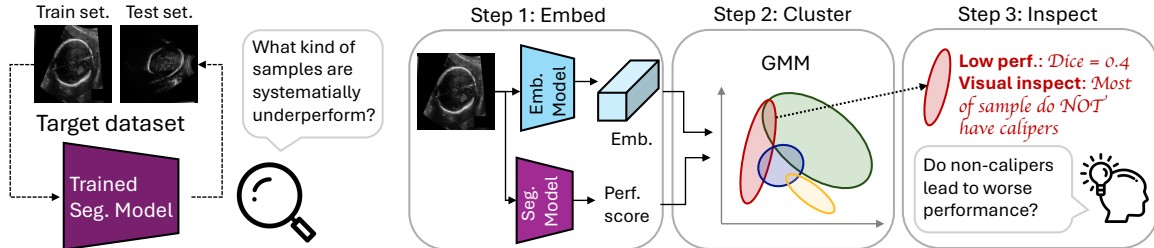

Figure 1: **Illustration of the SEG4SEG framework**: Three steps are included: (1) Embedding the image/annotation information; (2) Clustering using both image embedding and performance score; (3) Inspecting the problematic slices.

3. We propose a systematic taxonomy of failure modes in medical image segmentation and benchmark our SDM across four controlled-error settings and two real-world datasets, demonstrating SEG4SEG's effectiveness in uncovering diverse failure modes.

## 2. Problem Definition and Background

In this section, we formalize the problem of detecting systematic performance disparities in segmentation models and review existing SDMs developed for classification tasks, which our work builds upon.

### 2.1. Problem Set-up: Performance Disparities in Hidden Subgroups

We now formalize the learning setting and define how subgroup-related performance disparities manifest in segmentation models. Given a dataset $D = (x, y)_{i=1}^{N}$ and a hidden binary attribute $A$ (e.g. presence of caliper) , we assume that the marginal distribution of $A$ is consistent across the train, validation, and test splits[2], i.e. $P(A = 1|D_{train}) = P(A = 1|D_{val}) = P(A = 1|D_{test})$. We train a segmentation model $f$ using $D_{train}$ and perform model selection on $D_{val}$. Our goal is to do detect performance disparities with respect to $A$, i.e. $\mathbb{E}[\text{perf}(x) \mid A = 1] \neq \mathbb{E}[\text{perf}(x) \mid A = 0]$. Our goal is to develop a method that can detect such disparities from model outputs on unseen data, without requiring access to $A$ at inference time. In our experiments, we use and manipulate known $A$, to confirm the ability to surface $A$.

### 2.2. Slice Discovery Preliminaries: Concept and Motivations

SDMs aim to form clusters in which samples within each cluster are semantically similar, enabling the identification of slices that perform systematically worse. In practice, SDMs operate on feature representations of test samples, such as predictions, or foundation model embeddings. Let $z(x)$ denote the representation extracted from input $x$, which serves as the basis for clustering. The choice of $z(x)$ determines which types of failure modes the

---

2. This assumption follows prior work on SDMs and renders the SDM task more challenging compared to shortcut learning detection, where the distribution often differs in the test set.

SDMs can potentially uncover. For example, classification logits may reveal class-specific confusion, while encoder embeddings of the imaging captures image-level features.

Formally, let $\mathcal{D}_{\text{test}} = \{(x_i, y_i, A_i)\}_{i=1}^N$ denote the test set, where $A_i \in \{0, 1\}$ is a latent binary attribute indicating failure modes (e.g., annotation style, image quality). Given representations $\{z(x_i)\}_{i=1}^N$, an SDM produces a partition $S = \{s_1, s_2, \ldots, s_K\}$, where each slice $s_k$ consists of samples that are close in the representation space and exhibit similar performance. In our case, we aim to identify slices $s \in S$ that are predominantly composed of samples with $A = 1$, and whose average performance is lower compared to slices dominated by $A = 0$. Importantly, this clustering process is conducted in an unsupervised manner, without access to the underlying attribute $\{A_i\}$.

### 2.3. Existing SDMs for Classification: Capabilities and Limitations

**Pipeline Design.** SDMs for classification have evolved from simple clustering to sophisticated cross-modal frameworks.[3] Oakden-Rayner et al. (2020) pioneered the approach using $k$-means on pre-softmax features. DOMINO (Eyuboglu et al., 2022) popularized a workflow that combines foundation model embeddings (CLIP), clustering (GMM), and natural language interpretation. Subsequent work extended this framework: FACTS (Yenamandra et al., 2023) amplified clustering correlations, PlaneSpot (Plumb et al., 2023) improved on dimension reduction, and ViG-Bias (Marani et al., 2024) integrated visual explanations. An alternative paradigm, Spotlight (d'Eon et al., 2022), identifies contiguous low-performance regions without discrete clustering. However, all existing methods focus on classification, leaving segmentation unexplored.

**Evaluation.** Evaluating slice discovery remains challenging. Early work evaluated case-specific results through subgroup prevalence within slices (Oakden-Rayner et al., 2020; Olesen et al., 2024). DOMINO introduced precision-based metrics widely adopted in subsequent work including FACTS, but these metrics are limited to synthetic datasets with known bias attributes, motivating Bissoto et al. (2025) to propose improved metrics for real-world medical imaging settings. Building on this work, we further refine these metrics and introduce criteria to assess whether problematic slices are discovered.

## 3. A Taxonomy of Segmentation Failure Modes

We first systematically categorize documented segmentation failure modes as follows.

### 3.1. Shortcuts/Unwanted Correlation

Shortcut learning occurs when models exploit unwanted correlations between input features and targets instead of learning meaningful patterns (Geirhos et al., 2020; Lin et al., 2024). In segmentation, shortcuts can occur at both the sample level and pixel level:
**Sample-level shortcuts** arise from unwanted correlations between *sample-level features* and *segmentation performance*. One example is the presence of calipers in fetal ultrasound scans: models achieve better segmentation performance when calipers are present (Lin

---

3. Some approaches mitigate subgroup disparities without uncovering their causes (Jain et al., 2023; Kim et al., 2019; Sohoni et al., 2020). These fall outside our scope, as we focus on failure mode discovery.

et al., 2024). Including calipers in scans is common in clinical practice (Pu et al., 2022; Bano et al., 2021; Sun et al., 2022; Yang et al., 2020), yet their presence can mislead models into shortcut learning. We validate this failure mode in Sec. 5.1 (Case A).

**Pixel-level shortcuts** arise from unwanted correlations between *pixel locations* and *labels*. A representative example is spatial bias in skin lesion segmentation: Lin et al. (2024) demonstrated that when all training samples are center-cropped, models learn to systematically predict background labels for boundary pixels, regardless of content. Similar border artifacts appear in published models (Wang et al., 2023; Dai et al., 2022), where a U-Net/CA-Net failed to recognize lesions near image boundaries, suggesting widespread reliance on spatial priors rather than visual features. We validate this failure mode in Sec. 5.1 (Case B).

### 3.2. Label Noise

**Annotation styles** are often neglected in segmentation research, yet studies show that they can substantially affect model performance (Nichyporuk et al., 2022; Abhishek et al., 2024; Zhang et al., 2020; Zepf et al., 2023). Such variations encode the inherent subjectivity of labeling, driven by annotation protocols, rater expertise, and data pre-processing. We validate this failure mode in Sec. 5.1 (Case C).

**Annotation error**, inevitable in human-driven labeling, can introduce systematic failures for segmentation. Common manifestations are omission, inclusion errors and annotator-driven cognitive bias arising from boundary-related ambiguities (Vădineanu et al., 2022).

### 3.3. Underrepresentation

Class imbalance is a longstanding challenge in segmentation and a well-documented failure mode. Numerous remedies have been proposed (Li et al., 2020; Müller et al., 2022), particularly for structurally small anatomical regions such as retinal vessels (Fauzi et al., 2022). Underrepresentation also manifests at the sample level, where limited coverage of certain demographic groups (Puyol-Antón et al., 2021) or sub-disease categories leads to models that underperform on these minority slices.

### 3.4. Difficult Cases

Beyond the factors outlined above, certain subgroups may remain intrinsically difficult to segment due to image-dependent characteristics. Typical sources of increased complexity include degraded image quality (Jin et al., 2022), challenging anatomical topology such as curvilinear structures (Lin et al., 2023), and substantial variability in radiographic appearance (Heller et al., 2021). We validate this failure mode in Sec. 5.1 (Case D).

## 4. SEG4SEG: Slice Discovery Methods for Segmentation

Here, we propose SEG4SEG (Systematic Error Grounding for SEGmentation), which extends SDM to segmentation problems. The overall pipeline extends DOMINO (Eyuboglu et al., 2022) which was developed for classification.

### 4.1. Method Overview

SEG4SEG consists of three steps (see Fig. 1):

**Embedding the image space and performance metrics**[4]. Common approaches for image embedding include foundation models (e.g. CLIP) and latent representations from related pre-trained tasks. In this work, we use CLIP for image embedding and include both Dice score and per-image Positive Predictive Value as performance embeddings. To enable efficient clustering[5], we apply UMAP (McInnes et al., 2018) for dimension reduction of the image embeddings.

**Clustering embedding information**. Multiple clustering methods have been used in previous research, including k-means, HDBSCAN, and Gaussian Mixture Models (GMM). We use GMM in this work to enable flexible weighting between different variables. Specifically, we optimize

$$\ell(\phi) = \sum_{i=1}^{n} \log \sum_{j=1}^{|S|} P\left(S^{(j)} = 1\right) P\left(z(x_i) \mid S^{(j)} = 1\right) P\left(\text{perf}(\hat{y}_i, y_i) \mid S^{(j)} = 1\right)^{\gamma}, \quad (1)$$

where $z$ is the embedding model; $x_i$, $y_i$, and $\hat{y}_i$ denote the image, its annotation, and the prediction, respectively; and $\gamma$ is a weighting factor balancing the effect of performance embeddings and image embeddings.

**Inspecting the clustering results**. After clustering, we analyze the resulting clusters to identify potential issues in the dataset or model. This step typically requires additional annotations or analytical tools to characterize the discovered slices.

### 4.2. Representation Design for Segmentation Tasks

Since segmentation operates on pixel-level, we extend both the image and metrics representations to encode richer spatial information for more effective clustering. Specifically, we extend the representation as follows: **(a) Variants of image space embedding:** Apart from using embeddings from the original image $x_0$ as in the classification task, we explore alternative inputs by masking the image with either the ground-truth or predicted mask ($x_0 \cdot M_{\text{GT}}$ or $x_0 \cdot M_{\text{pred}}$), where non-foreground pixels are zeroed out, to encode additional mask information. **(b) Variants of performance space embedding**: Beyond commonly used overlap metrics (Dice score), we consider confusion-related metrics (Positive and negative predictive values: PPV, NPV) that distinguish FP and FN behaviors.

### 4.3. Evaluation Metrics for Slice Discovery

Following Bissoto et al. (2025), we evaluate SDM performance through the trade-off between performance disparities and slice purity. The goal is to discover slices that are pure regarding attribute $A$ and exhibit performance that deviates meaningfully from the overall population.

---

4. We treat both image features and performance metrics as embeddings, as both serve to compress information into a lower-dimensional space; despite the latter are not traditionally considered as embeddings.

5. High dimensionality leads to substantial inefficiency for most clustering methods (McInnes)

**Performance disparity.** In contrast to prior work, we measure performance disparity using the Omega Square ($\omega^2$) measure (Cohen, 1973), which quantifies the proportion of variance in performance explained by the partition $S$. Intuitively, $\omega^2$ measures how much better the partition is at separating high/low-performance samples compared to random grouping. We adopt $\omega^2$ instead of the difference between best and worst subgroup performance used by Bissoto et al. (2025), because the latter is sensitive to small cluster sizes and spurious results, whereas $\omega^2$ provides a more robust effect size measure.

**Purity.** Slice purity measures the homogeneity of a partition $S$ with respect to attribute $A$: $AP(S) = \frac{1}{|A|} \sum_{a \in A} \max_{s \in \hat{S}} \left( \frac{n_{s,a}}{n_s} \right)$, where $n_{s,a}$ denotes the number of samples belonging to attribute $a$ within slice $s$, and $n_s$ is the total number of samples in slice $s$. Here, $\hat{S}$ denotes the subset of all clusters $S$ satisfying $n_s > N_{\min}$. Intuitively, we exclude very small clusters when computing purity to avoid their noise-dominated behavior from inflating the metric.

The purity metric $AP$ does *not* penalize impure clusters if each attribute value $a \in A$ is concentrated in at least one slice. Both metrics aim to identify slices with high attribute purity and low performance. Impure clusters are acceptable, as representations may capture multiple data characteristics beyond target attributes.

### 4.4. Defining the success of finding the problematic slice

The two metrics introduced above assess, respectively, (i) whether an SDM can identify slices with low performance, and (ii) whether the SDM can isolate slices with high attribute purity. However, taken separately, these metrics do not directly answer the key question of interest: ***Does the SDM successfully locate the problematic slice(s)?***

To close this gap, we combine the two metrics and propose the following validation criterion to determine whether an SDM successfully identifies a problematic slice:

**Definition 1 (SDM Success Criterion)** An SDM is considered to successfully identify a problematic slice if *at least one* slice $s$ in the dataset satisfies all of the following conditions:

1. **High Purity:** $P_s(A = a_f) > p_\theta$. The proportion of samples in slice $s$ that share the failure-related attribute (noted as $a_f$) value exceeds the purity threshold.
2. **Low Performance:** $q(\text{perf}(s)) \leq q_\theta$. The quantile of the average performance of slice $s$ falls below the performance threshold.
3. **Sufficient Slice Size:** $n_s \geq N_{\min}$. The slice $s$ contains at least $N_{\min}$ samples.

If such a slice exists, it indicates that in a real-world setting where attribute labels are not always available, the SDM can locate low-performance slices, and the high purity in the slice might then help a human inspector to form a hypothesis as to the cause of the model failure. We set $p_\theta = 0.8$ to ensure clusters are sufficiently homogeneous (80% purity), $q_\theta = 0.4$ to capture the bottom 40% of clusters by performance, and $N_{min} = 2\%$ of the test set to enable detection of rare failure modes that may comprise only 5% of samples.

## 5. Experimental Design

As shown in Tab. 1, we select four representative cases from the failure mode taxonomy defined in Sec. 3, introduce the corresponding errors artificially – which are demostrated in

Table 1: Overview of failure modes and corresponding datasets.

| Case | Failure Mode | Real-world Case | Dataset | Failure Attribute ($a_f$) |
|------|--------------|-----------------|---------|---------------------------|
| A | Shortcut (sample level) | Calipers in ultrasound | HC18 | Non-caliper |
| B | Shortcut (pixel level) | Central cropping in skin lesions | ISIC2018 | Non-central masking |
| C | Annotation style | Boundary style in skin lesions | ISIC2018 | Polygon-style annotation |
| D | Difficult cases | Low-quality retinal images | FIVES | Low quality |

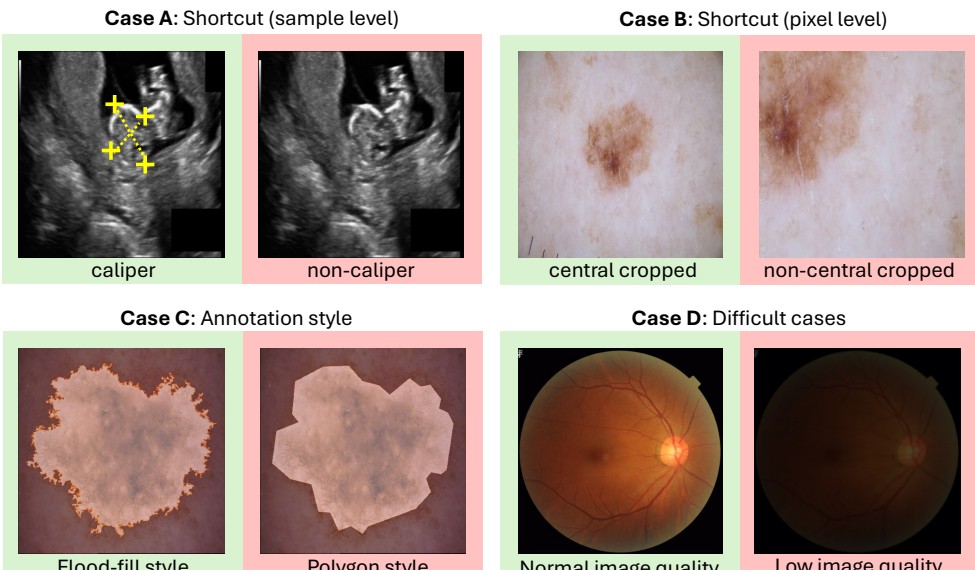

Figure 2: Illustration of failure modes across four experimental cases. Green backgrounds indicate samples that dominate the dataset, while red backgrounds highlight samples with systematic errors that we aim to identify and slice out using SDM methods. For demonstration purposes, we show the same sample with and without manipulation to amplify differences; in actual experiments, manipulated and unmanipulated versions of the same sample are never included in the same run. The added calipers in Case A are exaggerated for visual clarity.

Fig. 2 – and validate whether the SDM can reveal them. We also evaluate the framework on two real-world datasets that naturally contain annotation style inconsistencies (ISIC) and image quality variations (FIVES).

## 5.1. Experiment Set-up and Dataset Choices

**Case A – Sample-level Shortcuts: Calipers in Ultrasounds.** We apply our SEG4SEG to detect the caliper shortcut documented by Lin et al. (2024). We use the HC18 dataset (van den Heuvel et al., 2018) and introduce different error levels by generating artificial calipers from supplied segmentations.

**Case B – Pixel-level Shortcuts: Central Cropping in Skin Lesions.** We extend the analysis of Lin et al. (2024) on ISIC2018 by introducing synthetic cropping to reduce foreground-border correlations. For each cropping level $n$, we randomly select the top $n\%$ of samples, apply random crops at half the original image size, and discard crops that contain no lesion pixels.

**Case C – Annotation Styles: Skin Lesion.** Three distinct annotation styles are observed in ISIC2018: *flood-fill*, *jagged*, and *polygon* (Zepf et al., 2023). We conduct two experiments: (1) similar to Cases A and B, we introduce systematic errors by degrading *flood-fill* annotations into *polygon*-like ones at different levels, and (2) we use the original dataset with manually labeled annotation styles.

**Case D – Difficult Cases: Low Image Quality in Retinal Imaging.** FIVES is a retinal dataset annotated with three types of image quality degradation. Following Case C's design, we evaluate the SDM under two setups: (1) synthetic degradation by darkening ($\beta$=0.5 intensity scaling) and blurring (Gaussian $\theta$=2.0), and (2) the original dataset with existing quality annotations.

## 5.2. Implementation and Evaluation Details

**Evaluation pipeline.** Our evaluation follows a two-stage pipeline that simulates realistic failure discovery scenarios: (1) **Model training:** Train segmentation models on datasets with injected systematic errors (see Sec. 5.1 and Appendix B for error types and injection procedures). (2) **SDM-based discovery:** Apply SDM to the *test set* using the trained model predictions, original images, and ground-truth masks as inputs. Crucially, failure attribute labels are *not* provided to the SDM during slice discovery. Detection performance is then evaluated by comparing discovered slices against the known failure attributes.

**Segmentation model training.** All segmentation models for all cases are implemented using `segmentation_models_pytorch` (Iakubovskii, 2019) with a U-Net architecture and a ResNet-34 encoder, taking 3-channel RGB inputs. Models are trained for 50 epochs using Adam ($1 \times 10^{-4}$, batch size 32) with Binary Cross Entropy loss. Images are resized to $512 \times 512$ and augmented during training with random rotation, horizontal and vertical flips; all augmentations are disabled for validation and test splits. We use either the original train/val/test partitions provided with each dataset or a 64/16/20 split when original splits are unavailable (see Appendix B). Model selection is based on validation set performance.

**SDM configuration.** We perform 5 random runs for each experiment to estimate the *detection rate*, defined as the proportion of runs in which the problematic slice is successfully identified. Failure attributes are injected at 5%, 10%, and 20% prevalence. We applied CLIP to encode input images into 512-dimensional embeddings, and we applied UMAP for dimension reduction from the CLIP output, with output dimension of 8, the number of clusters are set as 10 for case A, C and D and 20 for case B. For representation extraction, we use CLIP to encode input images, followed by UMAP for dimensionality reduction to 8 dimensions. The number of clusters is set to 10 for Cases A, C, and D, and 20 for Case B. We sweep the weighting parameter $\gamma \in [10^{-3}, 10^3]$ (Eq. 1) and find that $\gamma = 10$ performs best across all settings; we report results using this value unless stated otherwise.

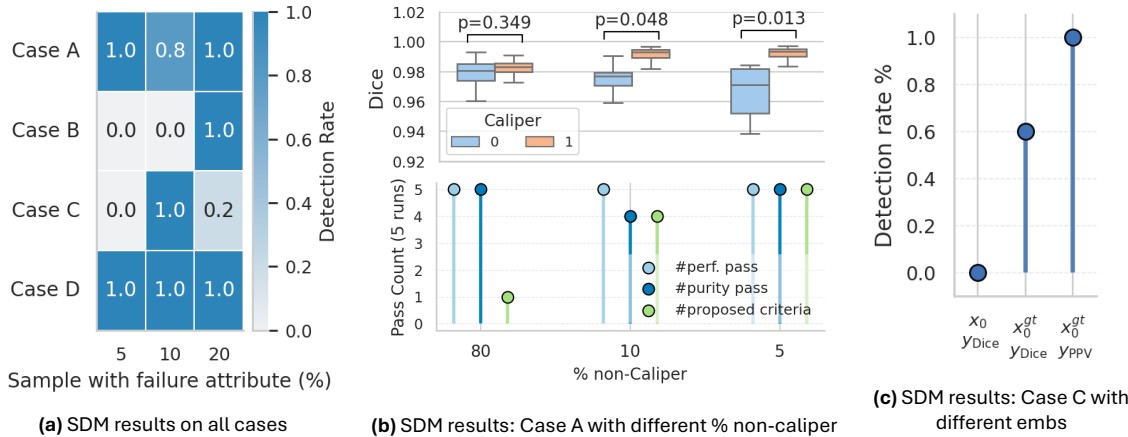

Figure 3: **SDM results.** (a) Proposed criteria evaluated over 5 random runs with varying failure-attribute proportions. (b) Case A. Top: the caliper vs. non-caliper performance gap is evident at prevalence 5% and 10% for non-caliper but negligible at 80%; Bottom: performance-pass/purity-pass shows no sensitivity to error levels, whereas the proposed criteria do. (c) Case C results with different embeddings. Statistical significance is assessed using independent two-sample t-tests.

## 6. SDM Results and Discussions

**SEG4SEG is able to discover problematic slices in segmentation.** Fig. 3a shows that our SDM effectively identifies performance-relevant problematic slices. In Cases A, C, and D, our SDM achieves high detection rates even at a prevalence as low as 5%. Case B with central cropping exhibits lower detection at low prevalence, likely because SEG4SEG operates at the sample level, whereas pixel–position shortcuts may require patch- or pixel-level grouping for precise identification.

**Purity and performance disparity alone do not capture slicing success.** Fig. 3b illustrates that neither purity nor performance disparity, whether considered independently or jointly, suffice to determine whether a slice is meaningfully identified. We define performance pass if $\omega^2 > 0.14$[6] and purity pass if $\text{Purity}(a_f) > 0.8$. However, even when caliper/non-caliper difference is negligible, both perf. pass and purity pass remain high, meaning that the low-performing slice is not necessarily the one driving the purity signal. This highlights a mismatch between these metrics and the actual reliability of slice discovery.

**The proposed evaluation criteria reflect whether a failure attribute affects model performance.** Fig. 3b shows that the proposed criteria increase their pass rates as the performance gap between caliper and non-caliper samples grows, indicating that the criteria align with the actual impact of the failure attribute.

---

6. Values for $\omega^2$ 0.14 indicates large effects. (Field, 2024)

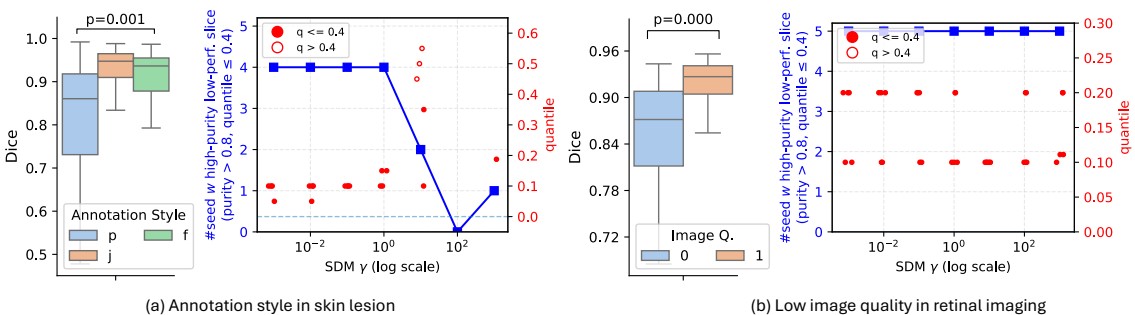

(a) Annotation style in skin lesion

(b) Low image quality in retinal imaging

Figure 4: **SEG4SEG uncovers failure modes in real-world datasets**: (a) annotation style in skin lesion segmentation, (b) low image quality in retinal imaging. Performance disparities are shown left, SDM results for different $\gamma$ are shown right. Statistical significance is assessed using independent two-sample t-tests.

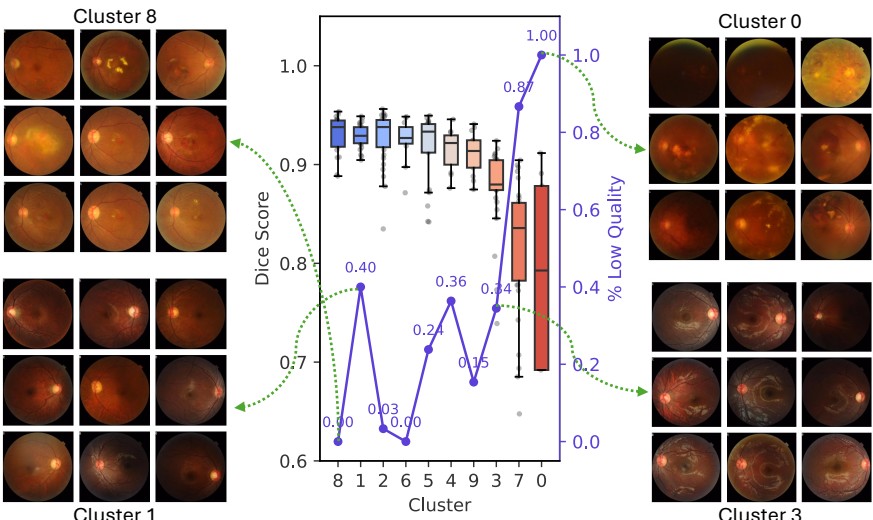

Figure 5: **Exemplary SDM result on the original FIVES dataset.** Center: Dice scores for the 10 clusters and the proportion of low-quality images in each cluster. Left and right: samples from selected clusters. The SDM identifies low-performance slices that contain a high proportion of low-quality images.

**Segmentation-specific embedding choices are critical for failure detection.** Case C illustrates the necessity of incorporating embedding variants (Sec. 4.2) as the error stems from annotation quality, not image appearance. As shown in Fig. 3c, using default embeddings (original images + Dice) fails to detect the degraded-annotation samples. Detection improves when the image embedding incorporates annotation masks, and improves further when replacing Dice with PPV, which is more sensitive to pixel-level false positives.

**SEG4SEG also works on real-world datasets.** Applying our SDM to the original ISIC2018 (Case B) and FIVES datasets (Case D) successfully identifies problematic samples arising from annotation style inconsistencies and low image quality, respectively. As shown in Fig. 4, the SDM highlights these failure modes by successfully passing the criteria, and the significant differences between the subgroups confirm the existence of these failures. Fig. 5 provides a detailed example of an SDM result for the low-quality retinal images. As illustrated in the center plot of Fig. 5, clusters with poorer performance exhibit a higher proportion of low-quality samples. Visualizing representative samples from each cluster further supports this trend: clusters with higher performance predominantly contain high-quality images, whereas cluster 0, which performs the worst, consists entirely of low-quality images. The SDM is also capable of capturing other meta-attributes: for example, one cluster groups together images with similar visual patterns, 93% of which correspond to cases without diagnosed eye disease.

## 7. Conclusion

In this work, we investigated whether SDMs can be applied to discovering failures modes in segmentation tasks. We have proposed a taxonomy of segmentation failure modes, selected representative test cases, and validated an adapted SDM (SEG4SEG) across four controlled-error settings and two real-world settings. SEG4SEG operates by clustering test samples based on their representations in an unsupervised manner, identifying slices where performance systematically degrades. Our results demonstrate that SEG4SEG is effective in revealing diverse failure modes in medical image segmentation and shows strong potential as a tool for systematic failure analysis.

### Acknowledgment

NW, AF and SB were partially funded by DTU Compute, the Technical University of Denmark; the Pioneer Centre for AI (DNRF grant nr P1); and the Novo Nordisk Foundation through the Center for Basic Machine Learning Research in Life Science (MLLS, grant NNF20OC0062606). This work was conducted during NW's external research stay, with the stay partially supported by Otto Mønsted foundation, IDAs og Berg-Nielsens Studie-og støttefond and travel scholarship from DTU. LMK and AB were supported by the Diabetes Center Berne. SS was supported by Deutsche Forschungsgemeinschaft (DFG) – EXC number 2064/1 – Project number 390727645, the Carl Zeiss Foundation in the project "Certification and Foundations of Safe Machine Learning Systems in Healthcare". The funding agencies had no influence on the writing of the manuscript nor on the decision to submit it for publication.

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

## Appendix A. Ablation Study

### A.1. Representation variants

The ablation study on representation variants discussed in Sec. 4.2, covering all four cases, is summarized in Tab. 2. For three out of four cases, the default setting, which used the original image $x_0$ as the image-space input and the Dice score as the performance-space representation, achieves the best detection performance. In contrast, for Case C, where the annotation style plays a critical role, using a groud truth annotation masked $x_0$ ($x_0 \cdot M_{\mathrm{GT}}$) improves the detection rate, particularly when combined with PPV-based metrics. Notably, masking $x_0$ using predicted annotations does not consistently improve performance across the four cases.

## Appendix B. Dataset Details

We summarize the datasets used in our experiments in Table 3. Additional details:

- For ISIC2018 that used in Case B and C: The original dataset contains 2594/100/1000 samples for training/validation/testing. Some samples are discarded in Case B (manipulated experiment) because when synthesizing the dataset by cropping, we discard cropped data where no lesion remains. For the manipulated failure mode of Case C, we select only the flood-fill annotations and degrade them to polygon style, which is why the dataset size is smaller (as there is no way to accurately perform the reverse conversion from polygon to flood-fill style).
- The annotation style labels in Case C are manually annotated by the authors and will be released with the code.
- For Case D with FIVES, the original dataset provides annotations for three low-quality issues: illumination and color distortion, blur, and low contrast. To simplify, we create a unified low-quality label (i.e., a sample is labeled as low quality if it exhibits any of these issues). For the manipulated Case D experiment, we collect all normal-quality samples and degrade them, resulting in fewer total samples in the manipulated variant compared to the real-world variant.

Table 2: **Ablation study on representation variants.** We evaluate variants of the image-space embedding, including the original image $x_0$, ground-truth-masked $x_0 \cdot M_{\mathrm{GT}}$, and prediction-masked $x_0 \cdot M_{\mathrm{pred}}$, as well as variants of the metric-space embedding, covering overlap-based metrics (Dice) and confusion-related metrics (PPV). Entries report the number of successful detections out of five random-seed runs under different failure-attribute ratios. The variant pair used in Fig. 3a is underlined.

| Image Space Variant | Perf. Space Variant | Detection Rate (out of 5 seeds) | | |
|---|---|---|---|---|
| | | Failure attribute ratio = | | |
| | | 5% | 10% | 20% |
| *Case A – Sample-level Shortcuts: Calipers in Ultrasounds* | | | | |
| $x_0$ | | 1.0 | 0.8 | 1.0 |
| $x_0 \cdot M_{\mathrm{GT}}$ | Dice | 1.0 | 1.0 | 1.0 |
| $x_0 \cdot M_{\mathrm{pred}}$ | | 1.0 | 1.0 | 0.8 |
| $x_0$ | | 1.0 | 0.8 | 0.4 |
| $x_0 \cdot M_{\mathrm{GT}}$ | PPV | 1.0 | 1.0 | 1.0 |
| $x_0 \cdot M_{\mathrm{pred}}$ | | 1.0 | 1.0 | 0.8 |
| *Case B – Pixel-level Shortcuts: Central Cropping in Skin Lesions* | | | | |
| $x_0$ | | 0.0 | 0.0 | 1.0 |
| $x_0 \cdot M_{\mathrm{GT}}$ | Dice | 0.0 | 0.0 | 0.8 |
| $x_0 \cdot M_{\mathrm{pred}}$ | | 0.0 | 0.0 | 0.2 |
| $x_0$ | | 0.4 | 0.2 | 1.0 |
| $x_0 \cdot M_{\mathrm{GT}}$ | PPV | 0.6 | 0.2 | 0.4 |
| $x_0 \cdot M_{\mathrm{pred}}$ | | 0.0 | 0.0 | 0.0 |
| *Case C – Annotation Styles: Skin Lesion* | | | | |
| $x_0$ | | 0.0 | 0.0 | 0.0 |
| $x_0 \cdot M_{\mathrm{GT}}$ | Dice | 0.0 | 0.6 | 0.4 |
| $x_0 \cdot M_{\mathrm{pred}}$ | | 0.0 | 0.0 | 0.0 |
| $x_0$ | | 0.0 | 0.0 | 0.0 |
| $x_0 \cdot M_{\mathrm{GT}}$ | PPV | 0.0 | 1.0 | 0.2 |
| $x_0 \cdot M_{\mathrm{pred}}$ | | 0.0 | 0.0 | 0.0 |
| *Case D – Difficult Cases: Low Image Quality in Retinal Imaging* | | | | |
| $x_0$ | | 1.0 | 1.0 | 1.0 |
| $x_0 \cdot M_{\mathrm{GT}}$ | Dice | 1.0 | 1.0 | 1.0 |
| $x_0 \cdot M_{\mathrm{pred}}$ | | 1.0 | 1.0 | 1.0 |
| $x_0$ | | 1.0 | 1.0 | 1.0 |
| $x_0 \cdot M_{\mathrm{GT}}$ | PPV | 1.0 | 1.0 | 1.0 |
| $x_0 \cdot M_{\mathrm{pred}}$ | | 1.0 | 1.0 | 1.0 |

Table 3: Dataset statistics and failure attributes. For manipulated variants, three imbalance ratios are evaluated: 95/5, 90/10, and 80/20 (dominant/problematic). For real-world variants, ratios reflect natural distributions.

| Case | Dataset | Size | Failure Attribute | Ratio (%) |
|------|---------|------|-------------------|-----------|
| *Manipulated Failure Modes* | | | | |
| A | HC18 | 999 | Caliper/Non-caliper | |
| B | ISIC2018 | 3673 | Central-cropped/Non-central-cropped | {95/5,90/10,80/20} |
| C | ISIC2018 | 731 | Floor-fill/Polygon | |
| D | FIVES | 592 | Normal-quality/Low-quality | |
| *Real-world Failure Modes* | | | | |
| C | ISIC2018 | 3694 | Flood-fill/Jagged/Polygon | train/val/test: {32.9/23.8/43.3, 9/65/26, 14.8/63.9/21.3} |
| D | FIVES | 800 | Normal-quality/Low-quality | train/test: {76.5/23.5, 66.5/33.5} |

