# OpenReview forum: "SEG4SEG: Identifying Systematic Failure Modes in Segmentation by Subgroup Discovery Methods"
_MIDL.io/2026/Conference — MIDL 2026 Poster_

### Official Review · Reviewer_FfSX · 2026-01-05

**Confidence:** 4
**Preliminary Rating:** 2
**Final Rating:** 4

**Summary:**

This paper describes a modification of a subgroup discovery method originally developed for classification, extending its applicability to segmentation tasks. The authors adopt a generic embedding + GMM framework, with the primary technical contribution focusing on the construction of embeddings suitable for segmentation tasks. The author validated their algorithms on 4 segmentation datasets with know shortcuts or subgroups for the effectiveness of their method.

**Strengths:**

Subgroup discovery is a clinically significant topic for improving the reliability of machine learning systems. The authors’ effort to propose and validate a solution that generalizes a classification-based method to segmentation tasks is well motivated and has the potential to benefit practical applications. If the method’s ability to identify small failure subgroups generalizes beyond the presented settings, it would be particularly commendable.

**Weaknesses:**

Technical:
1. In section 4.2, the author said "we explore alternative inputs by masking the image with either the ground-truth or predicted mask".
1.1 What is the exact implementation of this approach? Is that effectively zeroing-out the non-foreground pixel and reshape, or other methods?
1.2 Using the GT mask as a reference is not generalizable in practice if  the purpose if for the algorithm to be used for flagging unannotated images.
1.3 What is the performance of the model using predicted mask?
2.1 That author is not explicit about the configuration of the training set and the train-val-test splitting except the declaration in Sec 2.1. For my understanding (correct me if I am wrong), the author theoretically would use a training set containing all subgroups that may or may not trigger a failure, and validate and test on a set with all image subgroups but perfect annotations, and seeking to separate all subgroups in the test set, is that correct? (Case B, Case C)
3. The implementation of many implement is unclear. What pre-processing, augmentation, model, and optimizer is used for training the segmentation head that convert the CLIP embedding to the actual segmentation prediction?
4. What is the performance of alternative baseline?

Writing:
1. The writing of this manuscript needs structural improvement. Many subsections under section 2 and 3 can be made more clear by using tables to organize definitions and short explanation of terms.
2. I will suggest more effort in explicitly define the problem in unambiguous language, or even mathematical from, rather than the current problem-dataset form.
3. The author can be more explicit about the actual content and format of the embedding, both before and after the masking.

**Detailed Comments:**

See weakness.

For the language flow and taxonomy definitions, the author can consider the format and style in these two papers:
https://arxiv.org/abs/2006.12433
https://arxiv.org/abs/2305.06568

**Justification Of Final Rating:**

The authors have adequately addressed my concerns regarding the technical soundness and completeness of the manuscript by incorporating the requested new experimental results and previously missing technical details. Subject to further language revisions in the camera-ready version to improve readability, I recommend acceptance of the paper.

**Justification Of The Preliminary Rating:**

The paper proposes an extension of subgroup discovery to segmentation tasks, which is clinically significant. However, several key technical details are unclear, including the masking strategy, the use and performance of predicted versus ground-truth masks, dataset splitting, and the training configuration of the segmentation head, which limits reproducibility and interpretability. The reliance on ground-truth masks also raises concerns about practical applicability to unannotated data. Additionally, the manuscript would benefit from clearer structure, a more explicit problem formulation, and better organization of definitions and propositions.

**Questions To Address In The Rebuttal:**

1. Text readability, clarity in technical details, and structural flow.
2. Add baseline performance.
3. The performance of the method for cases where predicted mask, rather than group-truth mask is used (If applicable, depending on which usage case and application would the author claim)

---

> ### Author Response · Authors · 2026-01-25
> **Response to reviewer FfSX (Part 1/2)**
>
> We thank reviewer FfSX for the detailed and insightful feedback.
> In the following, we address their concerns (**W** as Weakness, **C** as detailed Comments, and **Q** as Questions):
>
> ----
> > **[W1 - method: implementation details]**
>
> * > “What is the exact implementation of this approach? Is that effectively zeroing-out the non-foreground pixel and reshape, or other methods?”
>
> Yes, we indeed zero out the non-foreground pixels. Thank you for pointing this out, e have now clarified this in Sec. 4.2 in the manuscript.
>
> * > “Using the GT mask as a reference is not generalizable in practice if the purpose if for the algorithm to be used for flagging unannotated images. ”
>
> Thank you for raising this question. The goal main goal of our work is not to develop a tool for day-to-day clinical practice but rather a debugging tool to identify systematic failure modes during the development process. In the development setting it is reasonable to assume that the GT is known, and thus it is justified to use the Dice score as a part of the input for clustering. We note that prior work on medical slice discovery, such as DOMINO (Eyuboglu et al., 2022) also assumed that the GT is available for slice discovery.
>
> * >“1.3 What is the performance of the model using predicted mask?”
>
> Thank you for this question. We have now added an ablation study with all representation variants for all cases in Appendix A.
>
> The results show that predicted mask embeddings are not as effective as GT mask embeddings for slice detection. We hypothesize that this is because the segmentation model struggles to learn two distinct annotation styles simultaneously, causing the styles to collapse in the predictions and thus losing the annotation style information that is critical for detecting problematic slices.
>
> -----
> > **[W2 - method ]** “2.1 That author is not explicit about the configuration of the training set and the train-val-test splitting except the declaration in Sec 2.1. For my understanding (correct me if I am wrong), the author theoretically would use a training set containing all subgroups that may or may not trigger a failure, and validate and test on a set with all image subgroups but perfect annotations, and seeking to separate all subgroups in the test set, is that correct? (Case B, Case C)”
>
> Yes, your understanding is correct, and this is indeed how we performed the experiments. We realized, however, that the experimental evaluation pipeline was not described sufficiently explicitly in the manuscript. To address this, we have now added a new paragraph (“Evaluation pipeline”) in Section 5.2 that clearly details the evaluation procedure. Thank you for pointing this out.
>
> -------
> > **[W3- method: implementation details / Q3]**  “The implementation of many implement is unclear. What pre-processing, augmentation, model, and optimizer is used for training the segmentation head that convert the CLIP embedding to the actual segmentation prediction?”
>
> We thank the reviewer for pointing this out. We added the detailed implementation details of the segmentation models in Sec 5.2 under "Segmentation model training".
>
> ----
> > **[W4 - method / Q2]** “What is the performance of alternative baseline?”
>
> As far as we know, we are the first to validate the SDM approach to segmentation model. Thus there are unfortunately no other baseline methods that we can compare to.
>
> ---
> > **[W5 - writing]** “The writing of this manuscript needs structural improvement. Many subsections under section 2 and 3 can be made more clear by using tables to organize definitions and short explanation of terms.”
>
> We are unsure how the reviewer  envisions the improved structure for Sections 2 and 3. We kindly ask the reviewer to clarify this point, and will do  our best to improve the clarity of the work for the camera ready version (given acceptance).
>
> In the meanwhile, we added a **new figure** to demonstrate the failure modes that we explore in this work (see **Fig. 2** in the revised manuscript). We believe that this figure also helps to better understand the systematic errors discussed in Section 3.

---

> ### Author Response · Authors · 2026-01-25
> **Response to reviewer FfSX (Part 2/2)**
>
> > **[W6 - writing]** “I will suggest more effort in explicitly define the problem in unambiguous language, or even mathematical from, rather than the current problem-dataset form.”
>
> Thank you for this suggestion. To clarify our structure: Section 2 provides the mathematical problem formulation (identifying hidden subgroups with performance disparities), while Section 3 taxonomizes specific segmentation failure modes that instantiate this general problem.
>
> We have also revised Section 2.2 to provide an explicit mathematical problem formulation.
>
> ---
> > **[W7 - writing]** “The author can be more explicit about the actual content and format of the embedding, both before and after the masking.”
>
> We are not entirely sure what aspect of the embeddings the reviewer was asking about, but here are the details:
> * CLIP produces 512-dimensional vectors, reduced to 8 dimensions via UMAP
> * Performance metrics (Dice or PPV) are computed as scalar features
> * Multivariate GMM is applied on the combined 8D UMAP features and performance metrics
> * When masking is applied, non-foreground pixels are zeroed out before CLIP encoding.
>
> We also revised the paper accordingly (mainly in Sec. 5.2) to clarify this. Please let us know if you were referring to something else.
>
>
> --------
>
> > **[C1/Q1]** “For the language flow and taxonomy definitions, the author can consider the format and style in these two papers: https://arxiv.org/abs/2006.12433 https://arxiv.org/abs/2305.06568”
>
> We thank the reviewer for these suggestions. Given the short duration of the rebuttal period we were not yet able to look into those papers in depth and integrate the structure into the revised manuscript. We will consider an improved structure based on the reviewer’s suggestions for the camera ready version (given acceptance).

---

> ### Comment · Reviewer_FfSX · 2026-01-31
>
> Thank the authors for addressing my concern regarding the technical soundness and reproducibility of the method. The current revision looks good to me from the technical perspective. To improve the readability of the camera-ready version, I suggest that the authors restructure the paper according to the following framework to achieve a clearer and more natural logical flow:
>
> Intro (Motivation, why this topic matters, a brief discussion about the current research landscape, potential application, your contribution in the current research landscape),
>
> Related Work (You can talk about the existing SDMs, for classification or segmentation),
>
> Preliminary (formal, self-contained problem definition, and all the failure modes you are currently including -- but leaving rooms pointing out others that you are not including)
>
> Method (expand on the validity of your algorithm/method and the experiment design, what is your hypothesis to be tested)
>
> Results
>
> Discussion and/or Conclusion

---

> > ### Author Response · Authors · 2026-01-31
> >
> > We are glad to hear that our reply addressed Reviewer FfSX's concerns regarding technical soundness and reproducibility! We also thank Reviewer FfSX for their suggestion on the paper structure. Due to the short time left in the rebuttal phase, we might not be able to do it now -- but we will definitely consider it for the further version given acceptance.

---

### Official Review · Reviewer_T1Wy · 2026-01-06

**Confidence:** 4
**Preliminary Rating:** 4
**Final Rating:** 5

**Summary:**

The authors present the first application of slice discovery for identifying failure modes in medical image segmentations tasks. They provide a taxonomy of failure modes in this context, and propose a metric that combines slice purity and performance for successful identification of problematic slices. The results show that the proposed SDM is effective on a range of modalities and failure types.

**Strengths:**

The paper is generally very well-written, with a good motivation and comprehensive background. The application of SDMs to segmentation tasks is novel, and the taxonomy presented is coherent and useful. The experiments showing multiple failure mode types/modalities are thorough and valuable. Fig 4 in particular is a really nice way of presenting both quantitative and qualitative results in a slice discovery context.

**Weaknesses:**

Due to the large amount of content included in this paper, I find that it is a bit confusing to follow some aspects, and some details/results are missing that would give a more complete picture of this work.

For example, some methodological details are missing/unclear wrt the results presented:
- How were the thresholds described in Sec 4.4 determined? If someone wanted to replicate this work/use this method, are they supposed to use the same thresholds, or was there a method for determining them empirically?
- “Detection rate” in Fig 2a is not really explicitly defined – I guess this is the proportion of times the success criteria is achieved over the five runs? This should be made clearer.
- What statistical tests were used in for Dice comparisons in Figs 2 and 3?
- Fig 3 is a bit confusing, and may need more detail for interpreting it. For example, is “# seed” the same as pass rate from Fig 2? Would be clearer if this was consistent or at least explained more.
- Which variant of image and performance embeddings were used for the real world datasets?

Also, some additional results (could be in an appendix) would make the paper more complete:
- The authors claim to use both predicted and GT masks as embedding variants, but only the GT variants are presented (Fig 2C). It would make the paper more complete to see these results as well (otherwise, why mention it in the methods?)
- It would be interesting to see the Fig 2C equivalent (incl. predicted mask embs) for all Cases.
- Dataset sizes as well as sizes of identified clusters would be nice to know.

**Detailed Comments:**

- Second and third sentences of paragraph 2 of the introduction have an awkward sentence structure. Would suggest changing “[...], and a successful classification [...]” to “[...], but a successful classification [...]”, and removing “however” from third sentence.
- First paragraph of Sec 6. should reference Fig 2A instead of just Fig 2.
- Could include a brief description of the slice discovery method in the conclusion.

**Justification Of Final Rating:**

The authors addressed all major reviewer comments, and have added additional clarifications to the manuscript which has improved it. This work represents a novel and valuable contribution to the community.

**Justification Of The Preliminary Rating:**

The paper is well written, novel, and includes extensive experiments. It would benefit from the additional results/clarifications mentioned above in order to be more reproducible and coherent to readers.

**Questions To Address In The Rebuttal:**

I have a few questions in addition to the ones brought up under “Weaknesses”:
- My understanding is that w^2 is a single measure of performance disparity across an entire set of discovered slices. I’m then confused about the “low performance” aspect of the SDM success criterion, where this appears to refer to performance of a singular slice. What is the performance metric used for this quantile evaluation then? And why not use the same performance metric thresholding when considering both detection rate and a “pass” (from fig 2B)?

- I’m a bit confused when comparing Figure 2(a) with 2(b). Why do we change from % caliper in (a) (assuming failure attribute = presence of caliper based on Sec 5.1) to % non-caliper? Also, for 20% caliper prevalence, (a) shows 1.0 detection rate, but (b) shows 0.2 pass count – why is there such a big difference here? (relates to the question above). I am also curious why the 20% non-caliper scenario does not show Dice performance disparities?

- How does using masked image representations impact the cropping and annotation style failure modes?

---

> ### Author Response · Authors · 2026-01-25
> **Response to reviewer T1Wy (Part 1/2)**
>
> We thank reviewer T1Wy for their insightful review. We are encouraged by the positive feedback on our paper’s motivation, the comprehensive background, novelty, taxonomy and experiments.
>
> In the following, we address the concerns (**W** as Weakness, **C** as detailed Comments, and **Q** as Questions):
>
> -----
> >  **[W1-method: Rationale under p_theta, q_theta, and N_min ]** “How were the thresholds described in Sec 4.4 determined? If someone wanted to replicate this work/use this method, are they supposed to use the same thresholds, or was there a method for determining them empirically?”
>
> **The values were chosen based on the following rationale:**
> * p_theta=0.8 (Purity threshold): This requires that 80% of samples within a cluster share the same attribute value, ensuring the cluster is sufficiently homogeneous to represent a meaningful failure mode.
> * q_theta​=0.4 (Performance quartile threshold): A cluster must rank in the bottom 40% of all clusters by performance to qualify as underperforming. This balances between being too permissive, i.e. capturing non-problematic clusters, and too restrictive, i.e. missing genuine failure modes. This threshold is set slightly below 0.5 to ensure we capture clusters that fall into the worse half while maintaining some selectivity.
> * N_min=2% of the test set (Minimum cluster size): The cluster must contain at least 2% of the test set samples. This threshold is motivated by our experimental design that failure attributes can comprise as little as 5% of total samples (with similar distributions across train/val/test splits). Setting N_min = 2% allows detection of failure modes even when they're rare. If we set N_min = 5%, only a cluster capturing all samples with the failure attribute would qualify, making the criteria overly restrictive.
>
> We revised our manuscript under sec 4.4 to make the rationale of the choice of the value more clear.
>
> **Recommended parameter value for the usage of other applications:**
> Our experimental results indicate that the method is robust across different representation variants, weighting factors (gamma), and datasets. Accordingly, we believe that the same or similar values are likely to also work for other applications.
>
> ----
> > **[W2-method]** “Detection rate” in Fig 2a is not really explicitly defined – I guess this is the proportion of times the success criteria is achieved over the five runs? This should be made clearer.”
>
> Yes, and thank you for pointing this out! We now added a description of detection rate under sec 5.2 in our manuscript to make it more clear.
>
> ---------
>
> > **[W3-method]** “What statistical tests were used in for Dice comparisons in Figs 2 and 3?”
>
> We used the independent two-sample t-test (scipy.stats.ttest_ind). We added this information to the captions of both figures. Thank you for the comment!
>
> ------
>
> > **[W4-method]** Which variant of image and performance embeddings were used for the real world datasets?
>
> We have 2 cases of the real world datasets: 1) annotation style with skin lesion dataset; 2) low image quality with retinal imaging. For 1), we used x_gt + y_ppv, which proved to work best on the manipulated dataset in Fig.2c ; For 2) we used x0 + y_dice (the default set-up).
>
> ------
> > **[W5/6-additonal result]** “The authors claim to use both predicted and GT masks as embedding variants, but only the GT variants are presented (Fig 2C). It would make the paper more complete to see these results as well (otherwise, why mention it in the methods?)” +  “It would be interesting to see the Fig 2C equivalent (incl. predicted mask embs) for all Cases.”
>
> Thank you for this suggestion. We have now added the complete ablation study of all representation variants for all cases in Appendix A.
>
> The results show that predicted mask embeddings are not as effective as GT mask embeddings for slice detection. We hypothesize this is because the segmentation model struggles to learn two distinct annotation styles simultaneously, causing the styles to collapse in the predictions and thus losing the annotation style information that is critical for detecting problematic slices.
>
> ------
>
> > **[W7-additonal result]** Dataset sizes as well as sizes of identified clusters would be nice to know.
>
> Thank you for this suggestion. We have added detailed dataset statistics, including sizes and preprocessing steps for all controlled experiments in Appendix B "Dataset Details".
>
> -----
> > **[C1/2/3]**
>
> Thank you very much, we have changed our manuscript accordingly.

---

> ### Author Response · Authors · 2026-01-25
> **Response to reviewer T1Wy (Part 2/2)**
>
> > **[Q1]** “My understanding is that w^2 is a single measure of performance disparity across an entire set of discovered slices. I’m then confused about the “low performance” aspect of the SDM success criterion, where this appears to refer to performance of a singular slice. What is the performance metric used for this quantile evaluation then? And why not use the same performance metric thresholding when considering both detection rate and a “pass” (from fig 2B)?”
>
> * “What is the performance metric used for this quantile evaluation then?”
>
> For the quantile evaluation, we use the Dice score to assess individual slice performance.
>
> * "Why not use the same performance metric thresholding when considering both detection rate and a 'pass' (from Fig 2B)?"
>
> We use different metrics for different purposes:
>   * **Why not use $\omega^2$ for the SDM criterion?** The criterion aims to identify at least one slice with both low performance and high purity. $\omega^2$measures variance across all slices simultaneously and cannot evaluate individual slice performance. Moreover, a high $\omega^2$ only indicates large overall variance but does not guarantee that the high-purity slice is the worst-performing one – which is critical for real-world applicability where experts must visually validate the worst-performing slices to identify failure causes.
>    * **Why not use Dice for performance disparity (Fig 2B)?** While Dice-based measurements have been used in previous work [Bissoto et al., 2025], we found they are highly sensitive to small cluster sizes. $\omega^2$  provides a more robust measure as it considers effect size across all slices.
>
> ------
> > **[Q2]** “I’m a bit confused when comparing Figure 2(a) with 2(b). Why do we change from % caliper in (a) (assuming failure attribute = presence of caliper based on Sec 5.1) to % non-caliper? Also, for 20% caliper prevalence, (a) shows 1.0 detection rate, but (b) shows 0.2 pass count – why is there such a big difference here? (relates to the question above). I am also curious why the 20% non-caliper scenario does not show Dice performance disparities?”
> * *“Why do we change from % caliper in (a) (assuming failure attribute = presence of caliper based on Sec 5.1) to % non-caliper? ”*
>
> We apologize for the confusion. In Case A, the failure attribute is NON-CALIPER (as specified in Table 1), not caliper presence. The logic is that the model learned to use calipers as a shortcut feature, which resulted in degraded performance on non-caliper cases. Therefore, non-caliper is the failure attribute we are tracking in this case.
>
> To clarify this distinction and avoid confusion, we have added a new Figure 2 that explicitly illustrates all four experimental cases, showing both the dominant samples and the samples with the failure attribute for each case. We hope this visual representation will help readers better understand the relationship between different attributes and failure modes across our experiments.
>
> * *“Also, for 20% caliper prevalence, (a) shows 1.0 detection rate, but (b) shows 0.2 pass count – why is there such a big difference here? (relates to the question above).”*
>
> The 0.2 pass count in Fig. 2b corresponds to 80% of cases NOT having calipers, not 20%. Fig. 3b shows that our proposed criteria makes more sense than using only a single measurement on the performance gap or purity score as it matches with whether there’s significance performance disparity between subgroups. We used 80%, 10% and 5% non-caliper on purpose.
>
> * *“ I am also curious why the 20% non-caliper scenario does not show Dice performance disparities?”*
>
> As mentioned above, the scenario results in 80% non-caliper cases rather than only 20%.
>
> -----
> > **[Q3]** “How does using masked image representations impact the cropping and annotation style failure modes?”
>
> In the default setting, we use x0 embedding and performance score (dice) as input for the clustering, which contains very little information about annotation style. By masking it with the GT/predicted mask, the embedding of the input encodes not only the pattern and texture of the image but also the annotation boundary, which helps to identify the different annotation styles.

---

> > ### Comment · Reviewer_T1Wy · 2026-01-27
> >
> > Thank you for the clarification on my questions, I think the paper has improved a lot.
> >
> > Wrt [Q3], my question was more out of curiosity about how the injected failure modes of Case B and Case C interact with the act of masking the image representation. I guess I'm just wondering if the authors think that these failure modes are uniquely impacted by masking vs. non-masking of the images, since those masks would (presumably) be impacted by cropping/annotation boundaries.
> >
> > Not something that needs to be addressed in the paper, just curious about hearing thoughts from the authors. :)

---

> > ### Author Response · Authors · 2026-01-28
> >
> > Thank you for the clarification and positive feedback!
> >
> > Regarding [Q3], that's a great question! We do believe that masking vs. non-masking could uniquely impact these failure modes, particularly because the masks inherently encode information about annotation boundaries. For Case C (different annotation styles), this relationship seems especially relevant, as the masked representations would directly capture boundary variations, which we observe the improvement of SDM result when change from x_0 to GT masked x_0) (Fig. 3c).

---

### Official Review · Reviewer_dz8D · 2026-01-10

**Confidence:** 4
**Preliminary Rating:** 5
**Final Rating:** 5

**Summary:**

The authors propose SEG4SEG, a slice discovery method (SDMs) to identify and analyze systematic failures in medical image segmentation. They first discuss the current state of SDMs and extend them to the segmentation domain by defining potential sources of errors and limitations of existing metrics. Results demonstrate the effectiveness of SEG4SEG across simulated settings (with known problematic slices) and real-world datasets. This work lays the foundation for future work in SDMs for segmentation.

**Strengths:**

- The problem setup is well-defined and recontextualized in the segmentation setting.
- The discussion of existing SDMs and their limitations in segmentation, leading toward defining potential types of failures a model may encounter, is a major strength. This not only clearly establishes novelty, but provides a baseline for future works in this field.
- The paper is well-organized and important findings are discussed in detail.
- The extension to real-world settings demonstrates the importance of SDMs to determine such problematic subsets. The inclusion of ablation study for $\gamma$ is a plus.

**Weaknesses:**

- Were $p_{\theta}$, $q_{\theta}$, and $N_{min}$ values arbitrarily chosen? If so, what was the justification for doing so? Additional clarity would help strengthen the paper.
- How are the segmentation models trained? If using prior methods, it would be helpful to include citations or brief summaries in the appendix.
- The authors should include a brief discussion on case D as all other cases are discussed in great detail. Did intentional degradation of image quality not impact GMM clustering? Or was the proposed metric good enough to still correctly identify problematic slices?

**Detailed Comments:**

- The paper includes some minor typos and grammatical errors (e.g., "Our goal is to do detect [...]. Our goal is to [...]" in section 2.1)
- I would recommend the authors include a figure visualizing the failures described in Table 1.
- In section 6 under " SEG4SEG is able to discover [...]", the figure reference should point to Fig. 2a rather than Fig. 2 as a whole to improve clarity.

**Justification Of Final Rating:**

I thank the authors for their detailed responses to my comments. All my concerns have been addressed, especially related to method clarity. In conclusion, this paper presents an interesting and novel extension of SDMs to segmentation and I recommend acceptance.

**Justification Of The Preliminary Rating:**

The paper is well-written, the experiments are well-defined and thorough, and the proposed method demonstrates its effectiveness and addresses the drawbacks of prior methods. In my opinion, this work lays the foundation for future work in this direction, namely defining sources of errors in segmentation models and extending existing literature in this domain. While certain aspects may benefit from additional clarity, I have to recommend this paper for acceptance.

**Questions To Address In The Rebuttal:**

Please see weaknesses and detailed comments.

---

> ### Author Response · Authors · 2026-01-25
>
> We thank reviewer dz8D’s positive feedback and valuable suggestions on our submission. It is encouraging to hear that the reviewer considers our work *to lay the foundation for future work in this direction* by *defining the systematic errors in segmentation* and *applying SDM to source the error*.
> We now address the concerns reviewer dz8D mentioned as below (**W** as Weakness, and **C** as detailed Comments):
>
> ----
> > **[W1: Rationale under p_theta, q_theta, and N_min ]** “Were p_theta, q_theta, and N_min values arbitrarily chosen? If so, what was the justification for doing so?”
>
> The values were empirically chosen, based on the following rationale:
> * p_theta=0.8 (Purity threshold): This requires that 80% of samples within a cluster share the same attribute value, ensuring the cluster is sufficiently homogeneous to represent a meaningful failure mode.
> * q_theta​=0.4 (Performance quartile threshold): A cluster must rank in the bottom 40% of all clusters by performance to qualify as underperforming. This balances the criterion between being too permissive, i.e. capturing non-problematic clusters, and too restrictive, i.e. missing genuine failure modes. This threshold is set slightly below 0.5 to ensure we capture clusters that fall into the worse half while maintaining some selectivity.
> * N_min=2% of the test set (Minimum cluster size): The cluster must contain at least 2% of the test set samples. This threshold is motivated by our experimental design that failure attributes can comprise as little as 5% of total samples (with similar distributions across train/val/test splits). Setting N_min = 2% allows detection of failure modes even when they are rare. If we set N_min = 5%, only a cluster capturing all samples with the failure attribute would qualify, making the criteria overly restrictive.
>
> We revised our manuscript under sec 4.4 to make the rationale of the choice of the value more clear.
>
> -------
>
> > **[W2]** “How are the segmentation models trained?”
>
> We thank the reviewer for pointing this out. We now added details about training the segmentation model in Section 5.2 under the new paragraph named "Segmentation model training.".
>
> ------
>
> > **[W3: more details about case D]** “The authors should include a brief discussion on case D as all other cases are discussed in great detail. Did intentional degradation of image quality not impact GMM clustering? Or was the proposed metric good enough to still correctly identify problematic slices?”
>
> We thank the reviewer for these comments. We note that case D is in fact already discussed in the Section titled “SEG4SEG also works on real-world datasets” with results shown in Fig. 4b and Fig.5, however we were referring to the case by the dataset name which may have led to confusion. We have clarified this in the updated manuscript.
>
> For Cases C and D, we conducted both controlled experiments with intentional image quality degradation to mimic failure modes (Fig. 3a) and real-world validation (Fig. 4, 5). As demonstrated in both figures, our metric successfully identified problematic slices in both scenarios, showing that GMM clustering remained robust despite the degraded image quality.
>
> We have added an Appendix section detailing the datasets used in all four cases and how we synthesized the datasets for controlled experiments.
>
> ------
>
> > **[C1]** “The paper includes some minor typos and grammatical errors (e.g., "Our goal is to do detect [...]. Our goal is to [...]" in section 2.1)”
>
> We performed a thorough proof-reading to eliminate the remaining typos and grammatical errors.
>
> ------
>
> > **[C2]** “I would recommend the authors include a figure visualizing the failures described in Table 1.”
>
> Thank you for this suggestion. We added a visualization of each of the failure modes in Fig. 2 in the revised manuscript.
>
> ----
>
> > **[C3]** “In section 6 under " SEG4SEG is able to discover [...]", the figure reference should point to Fig. 2a rather than Fig. 2 as a whole to improve clarity.”
>
> Thank you for pointing this out! We have revised the text accordingly (please note that due to the newly added Fig. 2this figure is now called “Fig.3a”).

---

### Author Rebuttal · Authors · 2026-01-25

**Rebuttal:**

We have revised our paper according to the reviewer's feedback. The changes are highlighted in orange.

**Supporting Material:**

/attachment/f71285afb7904f6fb5652ba84e8bf70e31d751ac.pdf

---

### Comment · Area_Chair_81Y7 · 2026-02-02

Dear Reviewers,
We kindly ask you to carefully read the authors’ rebuttals, if not done so already, before finalizing your review.
As we approach the final stage of the review process, please update your Final Rating for each assigned paper by navigating to:
“Edit” → “Official Review” → Final Rating. Kindly complete this update by February 1st, 2026 (23:59 AoE).
Thank you very much for your continued effort and valuable contributions.

---

### Meta-Review · Area_Chair_81Y7 · 2026-02-09

**Recommendation:** Accept (Oral)
**Confidence:** 4

**Metareview:**

The reviewers all agree that this is a strong and novel paper that makes a meaningful contribution by extending slice discovery methods (SDMs) to segmentation problems. Of particular note are the strong value of identifying systematic segmentation failures, the clarity of the motivation, and the thorough experiments. There were some concerns about the methodological clarity, but they all seem to have been addressed during the rebuttal phase.

I find that the paper offers a meaningful and novel extension of SDMs to segmentation and provides a foundation for future work in debugging segmentation models via subgroup discovery.

Ultimately, I believe this work is a valuable contribution to our field, and I would be excited to see it presented at MIDL.

---

### Decision · Program_Chairs · 2026-02-13

Accept (Poster)